# Does Ecological Agriculture Moderate the Relationship between Wine Tourism and Economic Performance? A Structural Equation Analysis Applied to the Ribera del Duero Wine Context

Rosana Fuentes-Fernández [1], Javier Martínez-Falcó [2,*], Eduardo Sánchez-García [2] and Bartolomé Marco-Lajara [2]

1   Business Management and Economics, University of Leon, 24001 Leon, Spain
2   Management Department, University of Alicante, 03690 San Vicente del Raspeig, Spain
*   Correspondence: javier.falco@ua.es

**Abstract:** The purpose of this research is to analyze the effect of wine tourism activity on economic performance in the wine context of Ribera del Duero (Spain), as well as the mediating effect of ecological agriculture on this link. To this end, a conceptual model is proposed based on the literature review carried out and contrasted through structural equation modelling (PLS-SEM) with data from 263 wineries, which in turn represent the population under study. The study results allow for us to empirically demonstrate the positive and significant relationship of wine tourism on performance, as well as the partial mediation of ecological agriculture in this relationship. The study thus contributes to the academic literature in a remarkable way given that, to our knowledge, there are no previous studies that have addressed the mediating role of ecological agriculture in the wine tourism–economic performance link. However, the research also suffers from certain limitations. In particular, given the relevance of the study, it is necessary to broaden its geographical scope so that, as a future line of research, it is proposed to contextualize the model proposed in the California wine industry, being able to subsequently establish similarities and differences in the Old and New World.

**Keywords:** ecological agriculture; wine tourism; economic performance; wine industry; Ribera del Duero; Spain

## 1. Introduction

The world of wine is attracting more and more interest, from learning about the vineyard, winemaking, and acquiring knowledge about grape varieties and their characteristics, as well as the bottling process. This interest has led to the emergence of enotourism, also known as wine tourism. Hall et al. [1], Getz [2], and Charters and Ali-Knight [3] agree that wine tourism is a sensory experience through various visitor attraction possibilities, including the experience of a lifestyle, the pleasure of tasting, or attending festivals.

This tourist alternative became so relevant and in demand that Europe established the definition of this term in the so-called European Charter of Wine Tourism [4], understanding it as the development of tourism, leisure, and free time activities dedicated to the discovery and cultural and oenological enjoyment of the vineyard, the wine, and its territory. Ten years later, the digital journal VINTUR [5] offered an official definition of wine tourism, conceiving it as "integration under the same thematic concept of the existing and potential tourist resources and services of interest in a wine-growing area".

Moreover, wine tourism is seen as an example of integral tourism that reflects the elements of the rural environment such as its folklore, its local gastronomy, or the customs of that geographical area [6]. In other words, the social, cultural and environmental history of a territory or the idiosyncrasy of its inhabitants can be defined in a global way as "the

wine landscape" [7]. In short, it is experiential tourism, which is based on the emotional relationship between the tourist and everything that surrounds the wine, creating a bond through the emotions experienced during the visit to the winery and/or its vineyards. The experiential visitor seeks to live the destination according to the experiences he/she wants to live.

Tourists who want to take part in this experience can do so from regions classified as Protected Designation of Origin (PDO) or Denomination of Origin Qualified (DOCa) [8]. Therefore, they are active visitors, they ask questions, they are interested, and even get involved—in some wineries they have, for example, the alternative of participating in the grape harvest, which would provide them with a unique experience. Alternatively, wine tourism might be seen as a model of sustainable and economic development. By promoting and improving the image of wine as a product, diversifying and seasonally adjusting tourist activity, it becomes one of the recent and potential drivers of the economy. Getz and Brown [9] consider that wine tourism is simultaneously a consumer behavior, a strategy to develop the geographical area and the wine market of that area, and a promotional opportunity for wineries to sell their products directly to consumers.

Since the 1990s, the diversification of a place's attractions and the offer of a greater variety of products throughout the year has encouraged alternative and/or complementary tourism to the sun and beach tourism typical of the months of July and August. Wine tourism would be one of the thematic tourism alternatives that meet these characteristics, by shifting preferences towards shorter and more frequent trips, avoiding the concentration of trips only for short periods and the tourist overcrowding that revives the well-known "tourism-phobia" (antipathy or aversion to tourism and tourists, especially when it becomes a mass phenomenon in an area).

On the contrary, wine and tourism form the perfect symbiosis to promote the socio-economic and environmental development of wine regions that are not overcrowded for tourism by creating jobs or generating wealth in rural areas through, among other alternatives, sustainable agriculture [10]. This is because wine tourism is a complementary element for rural development for three reasons: it increases tourist flows in that geographical area, it creates an important image of a quality tourist destination, and it serves to develop certain geographical areas [11,12]. In Spain, the first wine route was created in Cambados (Galicia) [13]. At present, the Association of Wine Cities (ACEVIN, for its acronym in Spanish), created in 1994 to establish the design and methodology necessary for different wine routes to become a reality, confirms that there are 34 certified routes in Spain by 2022, with three others already at an advanced stage: Uclés and Méntrida, in Castilla-La Mancha, and Txacolí, in the Basque Country.

During 2021, the Spanish wine route which received the highest number of visitors was the Calatayud Wine Route (Aragon), with 213,614 visitors, followed by the Ribera del Duero Wine Route (Castilla and Leon), with 197,145. The second wine-growing area received the approval of its PDO in 1982 by the Ministry of Agriculture, Fisheries and Food (MAPA, for its acronym in Spanish), and is the one selected for this study for two reasons: because it is one of the most important for both the quality of its wines and its volume of production, and because of its recent commitment to ecological agriculture. This route covers an area of around 21,000 hectares of vineyards and is the only route that includes municipalities in four Castilian provinces: Burgos, Valladolid, Soria, and Segovia. In fact, as Alonso et al. [14] (p. 112) point out, "Ribera del Duero is the largest PDO in Castilla and Leon, both in number of municipalities and registered hectares, as well as in production, number of wineries and winegrowers".

Along with wine tourism activity, wine tourists are interested in wineries whose production processes are more respectful of the environment—a concern that has reached viticulture. In this regard, Spain is the world's leading producer of organic wine. In 2021, 15% of the total vineyard surface area in this Iberian country was dedicated to organic production. In Castilla and Leon, this production increased by 21% in 2021 compared to the previous year. On a global scale, it accounts for 27% of the total area dedicated to

vine cultivation, with an average annual growth rate of 16%, according to data from the International Organization of Vine and Wine (OIV).

In this sense, the development of ecological agricultural practices can act as a pole of attraction for wine tourists with high environmental awareness, which in turn, has an impact on improving the economic performance of wineries. In addition, the production of organic wines derived from ecological agricultural practices developed by wineries can increase their differentiation, which can translate into greater economic performance. The present research aims precisely to analyze the mediating effect that ecological agricultural practices can have on the wine tourism–economic performance relationship, thus answering the following two Research Questions (RQs): (RQ1) does wine tourism positively affect the economic performance of wineries? (RQ2) do ecological agricultural practices mediate the wine tourism–economic performance relationship?

The study thus contributes to the academic literature and to wine industry professionals in a number of ways. First, the research advances the understanding of wine tourism in the Spanish wine industry, as well as the benefits of this activity. Second, to our knowledge, there are no previous studies that have addressed the mediating role of ecological agriculture in the wine tourism–economic performance link, so the research represents an advance in scientific knowledge. Third, the proposed model has not been previously proposed, which represents an opportunity to continue advancing the role that the development of wine tourism activities and ecological agriculture practices play in improving winery performance. Fourth, the study provides insight into the relationship between wine tourism and economic performance, which can be useful for winemakers who are considering developing wine tourism activities at their facilities. Fifth, through the results of this study, winemakers and winery environmental managers can learn about the role played by the development of ecological agricultural practices in improving winery profitability.

In order to achieve the two proposed research objectives, the study is divided into the following sections. After this brief introduction, Section 2 reviews the literature and sets out the research hypotheses to be tested. Section 3 presents the methodology, Section 4 the results of the study, Section 5 discusses these findings, Section 6 reflects on the theoretical and practical implications arising from the research, and finally, Section 7 presents the main conclusions.

## 2. Theoretical Background and Hypothesis Development

### 2.1. Wine Tourism and Economic Performance

Wine tourism has a particular impact on rural economies [15], favoring the sustainability of both wineries and the territory in which they are located [16]. This activity is considered a complementary means of job creation and wealth generation in rural areas for three reasons: it increases tourist flows in the areas where this type of tourism takes place, creates an important image of a quality tourist destination and serves to develop the socio-economic development of the wine-growing areas [17].

In addition to job creation, rural development can favor the deseasonalization of demand, as stated by numerous authors [15,18–20]. This rural incentive would be crucial, above all, for those rural areas where depopulation exists, understood as "the decrease in the number of inhabitants of a territory or nucleus" [21] (p. 2). This is the case in Castilla and Leon, a territory with deep-rooted farming traditions, where the populations of the four provinces of the Ribera del Duero PDO (Burgos, Segovia, Soria, and Valladolid) continue to reduce due to the rural exodus (see Scheme 1).

The effects of the depopulation of many rural areas of Spain have led Molino [22] to publish *Empty Spain. Journey through a country that never was*. The book indicates that there are two Spains: "There is an urban and European Spain [...], and an interior and depopulated Spain, which I have called Empty Spain". Following the definition of the Spanish Royal Academy (RAE, for its acronym in Spanish) empty means, "that is with fewer people than can be found in it" (RAE, definition 3).

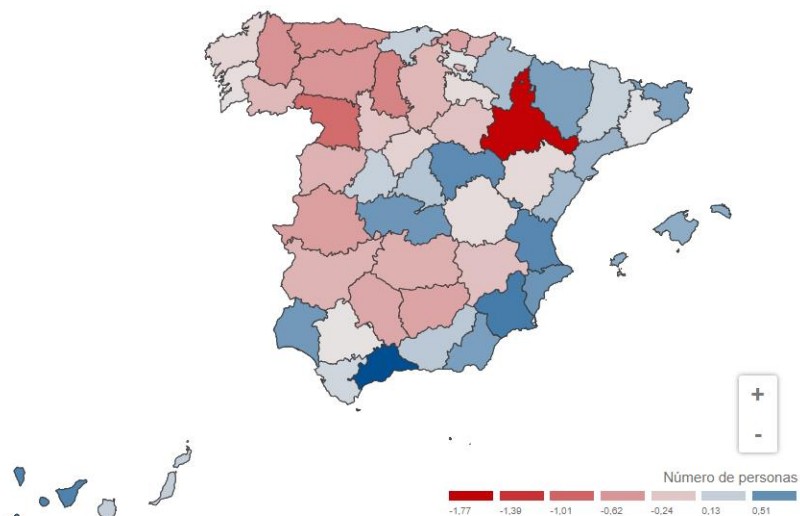

**Scheme 1.** Provinces where the population has decreased and increased in the last year. Source: Ep Data [23].

Rural development necessarily involves its balanced and self-sustainable revitalizations "based on its economic, social, and environmental potential through a regional policy and an integrated implementation of territorially based measures by participatory organizations" [24] (p. 48)—a premise that gives rise to the Sustainable Rural Development Programme 2010–2014 (Law 45/2007 of 13 December), which promotes a set of economic and public service guidelines to address the problems of depopulation, including the economic diversification of rural areas through wine tourism activities.

Wine tourism, therefore, is a good alternative to bring wealth, as well as to socio-economically revitalize wine-growing areas [25,26]. In fact, from an economic perspective, wine tourism can be understood as a distribution channel for the direct sale of wine in the winery [27], since it allows for (1) setting a lower price for wine than in other channels or a similar price but acquiring the margin kept by intermediaries; (2) obtaining instant liquidity compared to other means of wine distribution in which the entry of effect is delayed; (3) benefitting from possible up-selling and/or cross-selling; as well as (4) encouraging direct contact with customers, essential to guarantee future wine sales (generation of sentimental links with the brand and brand ambassadors, increased knowledge of the wine varieties offered by the winery, etc.).

Studies, such as the Town of Oliver the long-term community economic development function [28], confirm how wine tourism may affect the economic performance. One of the first achievements of Oliver's study was to recognize that wine tourism ranked as the most promising opportunity for Oliver and District Community Economic Development Society (ODCEDS). Martin and Williams [29] (p. 19) adopted a similar approach, speaking specifically about wine tourism development in British Columbia, they argue that "to sustain the value of these areas, municipal planners and managers need to work directly with their constituencies to develop clearer visions and policies concerning the role wine tourism will play in their regions". The same happens in both new and old European wine regions, where wine tourism "is experiencing significant development" [30] (p. 291).

Presenza et al. [31] (p. 46) suggest that "at the global level, the phenomenon of wine tourism is growing and is considered to be a driver for the economic and social development of many rural areas. These areas, although not traditionally characterized by a century -old tradition in the wine industry, are now playing an increasingly role in the current competitive scenario. One of the main development strategies implemented by wine producers is diversification in sectors either supporting the tourism sector or co-related to it". For this reason, and based on the literature review, we propose the following hypothesis:

**Hypothesis 1.** *Wine tourism affects the economic performance of wineries.*

*2.2. Wine Tourism and Ecological Agriculture*

There is a tourism called ecotourism which occurs in natural areas, "that is ecologically sustainable, that enables tourists to understand the environment which they are visiting, and which improves the socio-economic condition of local communities" [32]. This environmental awareness of visitors induced by a transformation of society's own values towards caring for the environment is also called "sustainable tourism" [33].

According to the World Tourism Organization, sustainable tourism is tourism that "takes full account of current and future economic, social and environmental impacts to meet the needs of visitors, the industry, the environment and host, communities." The Brundtland Report [34] goes further and defines this as meeting our present needs without jeopardizing future generations to meet theirs. This definition, applied to the wine industry, is similarly taken up by Gilinski et al. [35]—they point out that the main priority for practitioners in the wine industry is leaving the land in better conditions than the current ones for the next generation.

The interest in the conservation and protection of the ecosystem has changed consumer behavior towards certain products [36]. In the field of tourism, this change in habits has been reflected in the demand by visitors for regulated services that go hand in hand with the sustainable development of the environment so as not to damage it [37].

According to the OIV [38], the sustainable vitiviniculture is a "global strategy on the scale of the grape production and processing systems, incorporating at the same time the economic sustainability of structures and territories, producing quality products, considering requirements of precision in sustainable viticulture, risks to the environment, products safety, and consumer health and valuing of heritage, historical, cultural, ecological, and landscape aspects". Sustainable enological practices and respect for the vineyard, the wine, and its territory and its resources are clear examples of sustainable wine tourism, which can also be called 'enocotourism': a combination of enotourism and ecotourism.

Enocotourism is, therefore, inclusive and of high quality, as it puts quality and local tourism in the foreground, highlighting in addition the value of environmentally friendly agricultural practices. Marlowe et al. [39] demonstrated that wine tourism can act as a disseminator to explain the sustainable practices developed by the winery, making it possible to educate wine tourists about the importance of protecting the environment and the heritage that surrounds the wine-growing territory and the importance of the winery.

There appears to be a growing interest from producers and consumers, including tourist interest, in sustainable wine [40] and changing global wine consumer behavior [41]. Wine tourists value the preservation of the environment and the production of wines in the most sustainable way possible. In this line of thought, Bonn et al. [42] point to the growing number of consumers interested in consuming sustainable products. For the authors, this trend is justified by the growing sensitivity around environmental protection. This environmental attitude has been defined as "the set of beliefs and behavioral intentions that a person has with respect to activities or issues related to the environment" [43] (p. 31). Therefore, those wineries that develop environmentally friendly practices can attract a greater number of wine tourists, particularly those with high environmental awareness.

Another advantage of wine tourism is that it can be carried out on a small scale, which facilitates the protection of the landscape in the construction of the wineries themselves. The challenge is to preserve the natural resources and cultural integrity upon which sustainable wine tourism depends [44]. In this way, wine tourism can help to maintain and increase ecological agriculture practices, since it makes visible the effort made by wineries to develop these practices, as well as to capitalize on this effort by attracting new buyers.

To summarize, agritourism in general, where the wine tourism is included, offers the opportunity to provide "sustainable" or "green" tourism or "farm tourism" [45]. With

such reasoning, we agree that sustainable tourism plans merged with sustainable wine-production practices will lead to strong economic growth for wine tourism markets [46].

However, to the best of our knowledge, there are no previous studies that have attempted to analyze the effect of wine tourism on the development of organic agricultural practices. To overcome this research gap and based on the proposed literature, we propose the following hypothesis:

**Hypothesis 2.** *Wine tourism affects the ecological agriculture practices of wineries.*

### 2.3. Wine Tourism, Ecological Agriculture, and Wine Tourism

Wine tourism can enhance the value of ecological agricultural practices developed by the organization, in turn allowing for it to promote greater socio-environmental economic development of rural territories. Its professionalization is linked, above all, to the search for sustainability in the wine sector, given that the sustainable practices developed by wineries represent an added value in the visit offered to wine tourists [47].

The organic food market has been regarded as an emerging market [48]. Consumers, particularly in many industrialized countries, are aware of organic food [49]. Wine has been no exception, and this has been reflected in the rate of conversion of vineyards to green production which, according to the OIV, has increased by an average of 13% per year between 2005 and 2019. Almost a billion bottles of organic wine are already sold worldwide each year, twice as many as in 2013, according to a study carried out by IWSR [50]. For the OIV, this trend can be explained by current social demands, such as consumer health care and environmental protection.

According to Pomarici and Vecchio [51], there is a need for more in-depth studies involving consumer perceptions about ecological agriculture, as well as studies that investigate the correlation between purchasing decisions and attributes that motivate the consumption of organic wines. Whether consumer interests, attitudes, and perceptions of sustainability in general impact buying decisions is still highly disputed [52,53]. Some studies, however, confirm the direct link between wine tourism, economic performance, and sustainable agricultural practices. In this sense, Barber et al. [54] conducted a study with 300 consumers from the United States of America. The results showed that wine tourists, depending on their demographic profile and personality traits, were willing to pay more for organically produced wine and to pay a fee to taste wine or visit a wine region in order to protect the natural and cultural environment.

A winery that decides to go green will help the conservation of the territory. If it also hosts ecological tourism experiences, this translates into a contribution to regional development through the creation of new income opportunities for farmers by receiving visitors in the countryside. Cho et al. [55] claim that this tourism became an important tool for the economic development of rural localities. It is also beneficial for the wineries themselves as they can sell their own product during the wine tours. It has been suggested by Charters and O'Neill [56] that there are three major benefits that are derived from cellar door sales, that of distribution at a low marginal cost, the development of brand equity, and the chance to add value.

Several authors point out that these sustainable practices can also be used as a point of brand and product differentiation in a competitive market [57,58]. Ecological agriculture, therefore, supports differentiation, which is crucial for increasing productivity and competitiveness. Consequently, sustainability has developed into a priority in the wine supply chain [59,60]. In addition, consumers are willing to pay more for organic wine as demonstrated by several studies such as the one conducted by Mihailescu [61], which demonstrates the greater willingness to pay for organic wine, as well as to visit a winery that offers these wines in the South African wine context.

In spite of these advantages, wine regions do not automatically transform into eno-tourism destinations; a significant investment of time, money and effort is needed to

develop a successful wine tourism region. Williams and Dossa [62] (p. 26) argue that "conserving the natural resource base in wine regions is a product development function that requires the collaboration and sound planning of many partners". In this context, sustainability initiatives and measures could help firms implement proactive socio-environmental practices [63] and consequently improve their economic performance [64]. Furthermore, wine tourism generates business for wineries and other related products [65]. According to Williams [66], if the volume of wine tourism increases, the competitive positioning of wine tourism regions will also increase, making it a strategic issue.

The positive relationship is not only with wine tourism practices and economic performance, but these two variables may also be mediated by organic agricultural practices. Skinner [67] argues that as wine regions become increasingly involved in, or even dependent on, wine tourism, the need to sustain tourism as a source of economic resources is essential. Therefore, given the ability of organic agriculture practices to improve economic performance by increasing the differentiation of the wines offered, as well as attracting wine tourists, the following hypotheses are proposed to show the connection between organic agriculture and wine tourism with wineries' economic performance:

**Hypothesis 3.** *Ecological agriculture affects the economic performance of wineries.*

**Hypothesis 4.** *Ecological agriculture mediates the relationship between wine tourism and economic performance of wineries.*

Figure 1 shows a graphic representation of the theoretical model to be tested.

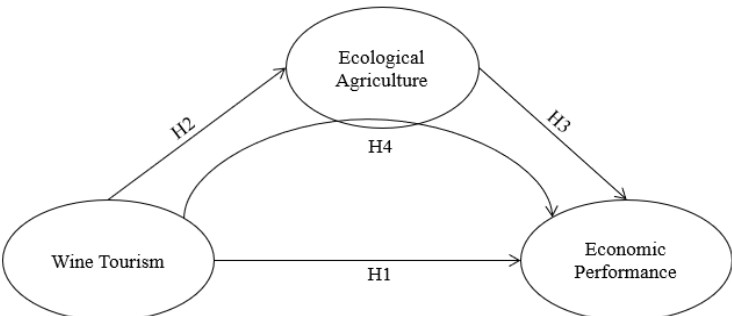

**Figure 1.** Theoretical model to be tested. Source: Own elaboration.

### 3. Methodology

To ensure adequate understanding of the methodological section, it has been divided into four blocks: (1) research context, (2) population and sample, (3) variables used and (4) analysis technique.

### *3.1. Research Context*

The study has been contextualized in the Spanish wine industry, in general, and in Ribera del Duero, in particular, for three main reasons. Firstly, Spain occupies a privileged position in the world wine industry, being the third largest wine producer in the world and the leading exporter in volume according to the latest data offered by the OIV. In this sense, within the Spanish wine territory, the role of Ribera del Duero stands out, being the PDO with the third largest area of registered hectares (43,042 ha), the third with the largest number of wineries (335), and the second with the largest wine commercialization (10.4 hl) according to the data offered by the MAPA for the 2020/2021 campaign. Secondly, the strategic importance of wine tourism in Spain is growing, being able to generate 91 million only in visits to museums and wineries attached to the wine routes in Spain [68]. In fact, within these wine routes, Ribera del Duero is among the top three with the greatest capacity to attract wine tourists and economic impact, generating more than EUR 12 million in visits to wineries and wine museums, representing 13.18% of the economic impact generated by

all the wine routes in Spain. Thirdly, the wine sub-sector is the most important after the meat one within the food and beverage sector in Spain, according to data provided by the Central Business Directorate of Spain. In the case of Ribera del Duero, the weight of the wine industry is accentuated, as it is the main element of identity and vertebration of the territory [69].

### 3.2. Population and Sample

The study population consisted of all active companies belonging to code 1102 of the National Classification of Economic Activities (CNAE, by its acronym in Spanish) and located in the Ribera del Duero region, which is made up of 19 municipalities in the province of Soria, 60 in the province of Burgos, 4 in the province of Segovia, and 20 in the province of Valladolid. After carrying out a search through the database of the Iberian System of Balance Sheet Analysis (SABI, by its acronym in Spanish), we were able to determine that there were 263 companies that met the triple condition of: (1) being active companies; (2) belonging to the CNAE code 1102 linked to winemaking; and (3) being located in the 103 municipalities that make up the Ribera del Duero region, which means that there were 263 wineries in our population. Likewise, in our research, the population coincides with the sample, i.e., the study sample is made up of 263 wineries; given this, to obtain the data, we made use of the economic–financial information provided by the SABI database, as well as through the wineries' websites, since through this database, we can find out the web address of each winery. Therefore, the units of analysis are the wineries, so that each observation represents one winery. The following subsection explains in more detail how the variables used in the study were measured and, therefore, which sources were used for their calculation.

### 3.3. Variables Used

The main variables used in the study were: wine tourism, ecological agriculture, and economic performance. Firstly, wine tourism has been considered as a dichotomous variable, taking the value 1 when the winery develops wine tourism activities and 0 when it does not develop such activities. In order to find out whether the wineries carried out wine tourism activities, the information provided by their corporate websites was used. Of the 263 wineries, 24 did not have a website, so they had to be contacted by telephone to obtain the information. Thus, of the 263 wineries analyzed, 124 carried out wine tourism activities. Secondly, the ecological agriculture variable was considered as a dichotomous variable with a value of 1 when the company sold organic, natural, or biodynamic wine and 0 when it did not sell any of these three types of wine. The commitment to ecological wine allows us to know that the winery is involved in ecological agriculture based on the optimal use of natural resources and, above all, that it discards the use of chemical or synthetic products. Thus, the winery was considered to be engaged in ecological farming practices if it offered at least one of the three types of wine (organic, natural, or biodynamic). As in the case of the wine tourism variable, the ecological agriculture variable was coded thanks to the information provided by the wineries' websites (except for the 24 wineries that did not have a website, which were contacted by telephone). It should be noted that of the 263 wineries, 117 offered ecological wine to the market and, therefore, developed ecological agriculture practices. Third, economic performance was conceived as a construct consisting of the following items: (1) the average of the last three years (2019–2021) of operating income, (2) the average of the last three years (2019–2021) of the fiscal year's results, (3) the average of the last three years (2019–2021) of financial profitability, and (4) the average of the last three years (2019–2021) of economic profitability. This information was obtained through the SABI database. Finally, PDO, size, and age were introduced as control variables. PDO was considered as a dichotomous variable, taking the value 1 when the company was adhered to at least one PDO and the value 0 when it was not adhered to any appellation. The size of each organization was measured according to the standards followed by the Organization for Economic Co-operation and Development [70]. With

respect to the age of the organization, this variable was incorporated by measuring the total number of years elapsed from the creation of the company until the time of the study (the year, 2022).

### *3.4. Analysis Technique*

The proposed theoretical model was tested by means of partial least squares structural equation modeling (PLS-SEM), using SmartPLS v 3.3.9 software. This technique allows for the analysis of a network of relationships between variables, some of which may be latent variables. This has intensified the use of this technique in the field of social sciences, in general, and in the field of management, in particular, since most of the variables in this field of research are not directly observable [71]. There are three reasons that have fundamentally motivated the use of this technique. First, the minimum sample size to be able to employ PLS-SEM is 100, fulfilling the minimum sample size requirement in the present study (n = 263). Second, direct and indirect relationships between constructs are established, with the use of PLS-SEM being recommended to analyze such a typology of relationships [72]. Third, the PLS-SEM technique has been previously employed to address strategic management aspects linked to the wine industry and the wine tourism subsector, highlighting the validity of the technique for conducting the present study [73–75].

### 4. Results

To report the results, we followed the indications of Hair et al. [72], who recommend presenting the results through three evaluations: (1) the evaluation of the global fit model, (2) the evaluation of the measurement model, and (3) the evaluation of the structural model.

First, regarding the evaluation of the global model, it should be noted that the model presents an adequate fit, since the Standardized Root Mean Square Residual (SRMSR) is $0.077 < 0.08$ [76], which means that it cannot be rejected and, therefore, is an adequate model to analyze the proposed relationships. Once the SRMR fit criterion was checked, we proceeded to verify if the unweighted least squares discrepancy (d_ULS) and the geodesic discrepancy (d_G) were within the confidence range after bootstrapping. Table 1 shows that the values of these indicators are below HI95 and HI99, thus fulfilling both requirements.

**Table 1.** Overall model fit.

|  | Value | HI95 | HI99 |
|:---:|:---:|:---:|:---:|
| SRMR | 0.077 | 0.083 | 0.091 |
| d_ULS | 0.598 | 0.643 | 0.704 |
| d_G | 0.619 | 0.667 | 0.762 |

Source: compiled by authors.

Table 2 shows the basic descriptive statistics consisting of the mean, maximum, and minimum values, as well as the standard deviation of each construct analyzed. As can be seen, the minimum and maximum values of the variables wine tourism (WT), ecological agriculture (EA), and PDO membership are 0 and 1, respectively, since they are dichotomous variables. Likewise, while the minimum number of workers in the wineries analyzed is 1 and the maximum 252, the minimum number of years is 1 and the maximum 226. Regarding economic performance, of the four indicators that make up the construct, the smallest value is 0 and the largest 198 (corresponding to the average operating income in the last three years in millions of euros). With regard to the results of the mean and the dispersion to the mean, it should be noted that the average number of workers and years is 12 and 14, respectively, with the ecological agriculture variable showing the greatest dispersion to the mean (1.231).

**Table 2.** Values of the mean, minimum value, maximum value, and standard deviation of the variables analyzed.

|  | Mean | Min | Max | Standard Deviation |
|---|---|---|---|---|
| WT | 0.471 | 0 | 1 | 1.196 |
| EA | 0.445 | 0 | 1 | 1.231 |
| EP | 4.873 | 0 | 198 | 1.043 |
| SIZE | 12.296 | 1 | 252 | 0.936 |
| AGE | 14.494 | 1 | 226 | 0.854 |
| PDO | 0.612 | 0 | 1 | 0.946 |

Source: compiled by authors.

Secondly, with regard to the evaluation of the measurement model, it should be noted that the indicators of the constructs analyzed meet the requirement of individual item reliability, as can be seen in Table 3, since their loadings are greater than 0.707, which means that the indicators present adequate levels of reliability [72]. Similarly, all the constructs meet the internal consistency reliability criterion, since both the composite reliability (pc) and Cronbach's alpha have values above 0.8, as well as the convergent validity criterion, since the values of the Average Variance Extracted (AVE) are above 0.5 [77], thus showing that the constructs are able to explain more than half of the variance of their indicators. As can also be seen, the composite reliability values, Cronbach's Alpha and AVE, have a value of 1 for both the wine tourism and the ecological agriculture variable. This is due to the fact that both constructs are composed of a single indicator. Table 4 shows the discriminant validity test of the variables analyzed according to the Heterotrait-Monotrait criterion (HTMT). As can be seen, all the values of the constructs are clearly lower than 0.85, so that each construct measures different realities.

**Table 3.** Measurement model analysis: external loadings, construct reliability, and convergent validity.

| Construct/Items | Outer Loadings | Rho_c (Pc) | Cronbach's Alpha | AVE |
|---|---|---|---|---|
| Wine Tourism |  | 1.000 | 1.000 | 1.000 |
| WT1 | 1.000 |  |  |  |
| Ecological Agriculture |  | 1.000 | 1.000 | 1.000 |
| EA 1 | 1.000 |  |  |  |
| Economic Performance |  | 0.902 | 0.867 | 0.731 |
| EP 1 | 0.716 |  |  |  |
| EP 2 | 0.779 |  |  |  |
| EP 3 | 0.748 |  |  |  |
| EP 4 | 0.792 |  |  |  |

Source: compiled by authors.

**Table 4.** Discriminant validity analysis based on the HTMT criterion.

|      | WT    | EA    | EP    | SIZE  | AGE   | PDO |
|------|-------|-------|-------|-------|-------|-----|
| WT   |       |       |       |       |       |     |
| EA   | 0.034 |       |       |       |       |     |
| EP   | 0.021 | 0.168 |       |       |       |     |
| SIZE | 0.048 | 0.017 | 0.182 |       |       |     |
| AGE  | 0.069 | 0.029 | 0.042 | 0.021 |       |     |
| PDO  | 0.045 | 0.284 | 0.246 | 0.268 | 0.125 |     |

Source: compiled by authors.

Third, regarding the evaluation of the structural model, before proceeding with the structural model analysis, the possible presence of collinearity problems in the structural model was examined. According to the indications of Hair et al. [72], there are indications of quality when the Variance Inflation Factor (VIF) is greater than 5 (VIF > 5). That is, values greater than 5 in the endogenous constructs imply critical levels of multicollinearity. In this sense, the VIF values obtained in the present investigation do not exceed the pre-established maximum threshold for any of the variables (see Table 5).

**Table 5.** Analysis of collinearity in the model through VIF values.

|      | WT | EP    | EA    | SIZE | AGE | PDO |
|------|----|-------|-------|------|-----|-----|
| WT   |    | 1.249 | 1.624 |      |     |     |
| EP   |    |       |       |      |     |     |
| EA   |    | 1.531 |       |      |     |     |
| SIZE |    | 1.143 |       |      |     |     |
| AGE  |    | 1.002 |       |      |     |     |
| PDO  |    | 1.611 |       |      |     |     |

Source: compiled by authors.

Figure 2 shows the R-Squared and β-squared results based on a bootstrap test with 5000 subsamples. The results of the research reveal that wine tourism has a positive and statistically significant effect on economic performance (0.340); this activity also has a positive and significant effect on ecological agriculture (0.249), and this in turn has a positive effect on economic performance (0.326). This implies that the ecological variable construct partially mediates the wine tourism–economic performance relationship, given that both the direct (0.340) and indirect (0.081) effects are positive and significant, with a total effect of wine tourism on economic performance of 0.421 (see Table 6). Therefore, the four hypotheses put forward can be accepted. Table 7 shows the degree to which an exogenous construct allows for explaining a given endogenous one in terms of $R^2$, i.e., it shows the results of the effect sizes ($f^2$). As can be seen, the most representative $f^2$ values correspond to the effect of wine tourism in explaining the economic performance variable (0.362), being also, as explained above, the relationship with the highest path coefficient. Likewise, to analyze the quality of the model, the Geisser test ($Q^2$) was performed, which must present estimated values greater than 0 ($Q^2 > 0$). As can be observed through Table 8, the values reflect an average predictive relevance of the model, since the $Q^2$ values are higher than 0.25 [71]. Finally, regarding the control variables, the results reveal that while size and PDO membership have a positive relationship on economic performance, winery age has a negative relationship on this variable. However, none of the three relationships are significant, so the results cannot be extrapolated to the study population.

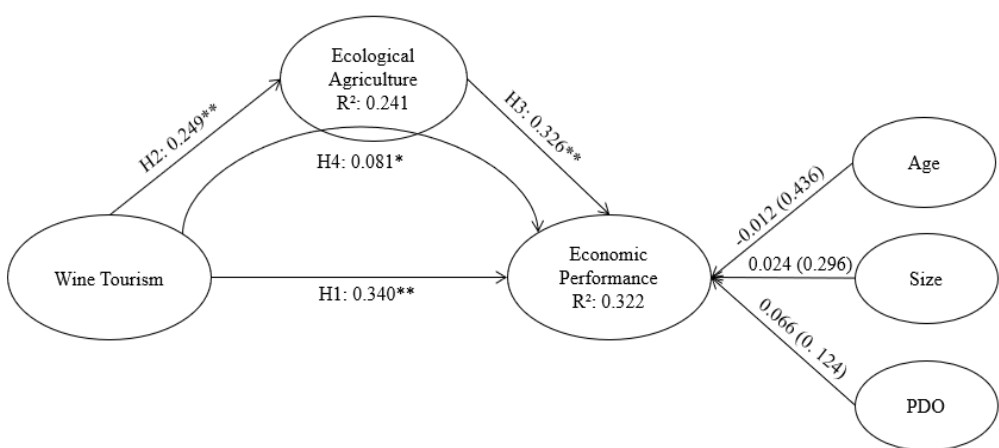

**Figure 2.** Theoretical model with R-squared, path coefficients (β), and significance. Source: compiled by authors.

**Table 6.** Results of the structural model for the mediation model.

| Direct Effects | Path Coefficient | t-Value | *p*-Value | 95% BCCI | Hypothesis Supported |
|---|---|---|---|---|---|
| WT -> EP | 0.340 | 4.635 | 0.000 ** | [0.295; 0.451] | H1 supported |
| WT -> EA | 0.249 | 10.417 | 0.000 ** | [0.183; 0.428] | H2 supported |
| EA -> EP | 0.326 | 3.737 | 0.000 ** | [0.212; 0.431] | H3 supported |
| Indirect Effects | Path Coefficient | t-Value | *p*-Value | 95% BCCI | Hypothesis supported |
| WT -> EA -> EP | 0.081 | 2.572 | 0.003 * | [0.054; 0.206] | H4 supported |

Notes: BCCI = Bias Corrected Confidence Intervals; ** $p < 0.001$ * $p < 0.005$. Source: compiled by authors.

**Table 7.** Effect sizes ($f^2$) of the analyzed variables.

| | EP | EA |
|---|---|---|
| WT | 0.362 | 0.112 |
| EA | 0.073 | |
| SIZE | 0.094 | |
| AGE | 0.088 | |
| PDO | 0.021 | |

Source: compiled by authors.

**Table 8.** Construct cross-validated redundancy.

| | SSO | SSE | $Q^2$ (=1-SSE/SSO) |
|---|---|---|---|
| EA | 1416 | 1156.322 | 0.308 |
| WT | 1212 | 1212 | |
| EP | 1030 | 645.321 | 0.278 |
| SIZE | 202 | 202 | |
| AGE | 202 | 202 | |
| PDO | 202 | 202 | |

Source: compiled by authors.

## 5. Discussion

The link between wine tourism activities and economic performance can be understood through the economic contribution of this type of tourism to the winery, since it

allows for (1) acquiring the margin kept by intermediaries; (2) obtaining instant liquidity, unlike other channels in which cash flow is delayed; (3) promoting cross-selling and incremental sales; as well as (4) generating an emotional bond with the customer, so that they can become brand ambassadors. Therefore, the development of wine tourism activities can increase the economic performance of wineries both by increasing their direct wine sales and by improving their commercial skills. It should also be noted that this activity plays a fundamental role for small wineries, as these are the ones that have the greatest difficulty in accessing the large wine distribution channels. The results regarding the wine tourism–economic performance link are in line with previous research in the field of wine tourism, such as those of Canziani et al. [78], Smyczek et al. [79], and Sun and Drakeman [27].

Regarding the link between wine tourism and the development of ecological agriculture practices, this relationship can be explained through the increase in the stock of ecological knowledge derived from the wine tourism activity, since in order to transmit the environmental practices carried out by the winery, the wine tourism managers can interact and be in contact with other members of the winery (winemakers, quality, and environmental managers, etc.), as well as attend training courses of an ecological nature, which allows for improving the green knowledge of the wine tourism managers. Likewise, environmental questions and suggestions from wine tourists can also increase the stock of green knowledge of the workers in charge of carrying out the wine tourism activity in the winery. In this way, the increased environmental knowledge of the members of the winery can lead to the improvement of ecological agriculture practices, which can serve as a pole of attraction for wine tourists with a high awareness of the protection and care of the environment. Therefore, the link between wine tourism and ecological agriculture can occur both by increasing the green knowledge of workers and by the winery's willingness to attract tourists with high environmental awareness, which, in turn, can lead to an increase in its economic performance as a result of the greater ability to attract wine tourists, as well as the greater differentiation of the winery by being able to offer ecological wines to the market. This is in line with the research carried out by Grimstad and Burgess [80], Karagiannis and Metaxas [81], and Trigo and Silva [82], who consider that wine tourism can enhance the value of the ecological agricultural activities carried out by the winery, and can also serve to attract wine tourists who are aware of the need to protect the environment.

Regarding the positive link between organic agricultural practices and the economic performance of wineries, there is recent research that coincides with the findings of the study, such as that of Merot et al. [83], Ingrassia et al. [84], and Katunar et al. [85], who demonstrate the capacity of these practices to improve the economic performance of wineries, since they favor the green organizational image, enable the attraction of new wine tourists, and allow for them to offer organic wines to the market. The results regarding the mediating effect of ecological agriculture in the wine tourism–economic performance relationship are pioneering, since, to our knowledge, there are no previous studies that have addressed the mediating effect of ecological agriculture in the study of the main relationship.

## 6. Theoretical and Practical Implications

Several practical and theoretical implications are derived from the study. Regarding the theoretical implications, the research contributes to elucidating the economic benefits of wine tourism activity in wineries by empirically demonstrating the positive relationship between wine tourism and the economic performance of Ribera del Duero wineries. Furthermore, to the best of our knowledge, there are no previous studies that have empirically analyzed the mediating role of the development of ecological agriculture practices in the wine tourism–economic performance link, which represents an important advance in scientific knowledge, improving the understanding of the benefits that can be derived from the development of wine tourism activities.

In terms of practical implications, the study can be of great use to winemakers who are considering the implementation of wine tourism practices in their facilities, since it can demonstrate its impact on the economic profitability of the winery, as well as on the devel-

opment of ecological agriculture practices, such as the use of organic vineyards, fertilization with organic materials such as compost, green manure or harvest residues, increasing diversity in and around the vineyard with different plants that stimulate diversity and promote favorable climatic conditions for the development of the vineyard, and controlling yields to obtain quality fruit. These practices, in turn, can lead to improved economic performance for the winery, as a result of the differentiation that the development of these practices can bring with respect to other wineries in the market. In fact, this differentiation can be capitalized through the wine tourism activities developed by the winery, which can lead to the enhancement of this type of ecological practices in the long term.

## 7. Conclusions

This research empirically demonstrates the positive and significant relationship between wine tourism and economic performance, as well as the mediating effect of ecological agriculture in this relationship. For this reason, the study may be of great interest to both academics and wine industry professionals who wish to understand the economic benefits that wine tourism activity can bring, as well as the mediating role of ecological agriculture in this link.

Wine tourism represents a crucial activity for wineries to improve their competitiveness (by increasing the direct sales of wine in the winery) and their organizational innovation processes (as it involves product innovation), as well as to boost the territorial development of a given wine region (by encouraging the creation and retention of employment in the territories where the wineries are located). Additionally, as demonstrated in this research, wine tourism activity can catalyze ecological farming practices in wineries. Thus, the study allows us to answer the research questions posed, since (RQ1) a positive and significant effect has been demonstrated between the development of wine tourism activities and economic performance, as well as (RQ2) the partial mediation of organic agriculture in this link.

The research, therefore, allows us to understand the meaning and significance between the variables wine tourism, organic agriculture, and economic performance. It highlights the catalytic role of wine tourism in improving the profitability of wineries by improving the direct, cross, and incremental sales of wine on the premises, as well as favoring environmentally friendly practices, such as organic agriculture. These practices, in turn, can act as a pole of attraction for wine tourists, improve the image and corporate reputation, as well as enable the production of organic wine, which could lead to an improvement in the economic performance of the wineries.

Despite the relevant contributions presented in this study, it is important to point out that the research suffers from certain limitations. In particular, given the relevance of the study, it is necessary to extend the geographical scope to other New World wine regions, thus, being able to establish similarities and differences in the model proposed for the New and Old-World wine contexts. In addition, the study has been nourished only by secondary data. To overcome both limitations, as a future line of research, we intend to extend the analysis to Californian wineries in order to establish similarities and differences in the proposed model, formulating a questionnaire to obtain primary information.

**Author Contributions:** Conceptualization, R.F.-F. and J.M.-F.; methodology, B.M.-L.; software, E.S.-G.; validation, R.F.-F., B.M.-L. and J.M.-F.; formal analysis, E.S.-G.; investigation, R.F.-F.; resources, B.M.-L.; data curation, J.M.-F.; writing—original draft preparation, E.S.-G.; writing—review and editing, R.F.-F.; visualization, B.M.-L.; supervision, E.S.-G.; project administration, J.M.-F. All authors have read and agreed to the published version of the manuscript.

**Funding:** This research received no external funding.

**Institutional Review Board Statement:** The present study did not involve humans or animals.

**Informed Consent Statement:** Not applicable.

**Data Availability Statement:** The datasets used and analyzed during the current study are available from the corresponding author on reasonable request.

**Conflicts of Interest:** The authors declare no conflict of interest.

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
