# Peer review of "Does Ecological Agriculture Moderate the Relationship between Wine Tourism and Economic Performance? A Structural Equation Analysis Applied to the Ribera del Duero Wine Context"

_agriculture, doi:10.3390/agriculture12122143_

Round 1

Reviewer 1 Report

I find that manuscript interesting and well-written. The structure of the article is correct, the research was conducted in a reliable manner and the conclusions are logical.

Although I suggest not listing research questions in the Introduction part. These research questions coincide with the scope of hypotheses H1 and H2.In the Conclusion part, there should be a clear statement to each hypothesis with the information if it was verified positively or not.

I am happy to read that the Authors plan to continue research in the California region. The result of the comparison between these two regions may be interesting.

Author Response

“I find that manuscript interesting and well-written. The structure of the article is correct, the research was conducted in a reliable manner and the conclusions are logical.”

Thank you very much for your kind words, it means a lot to us.

“Although I suggest not listing research questions in the Introduction part. These research questions coincide with the scope of hypotheses H1 and H2.In the Conclusion part, there should be a clear statement to each hypothesis with the information if it was verified positively or not.”

Thank you very much for your suggestion, based on your recommendation we have included in the conclusions section the two research questions derived from the results of the study.

“I am happy to read that the Authors plan to continue research in the California region. The result of the comparison between these two regions may be interesting.”

We are currently in the process of collecting data. We are sure that we will bring new knowledge to the field of wine tourism in the Old and New World. Thank you very much for your comment.

“Thank you for giving me the opportunity to review this paper. However, I have shared my viewpoints.”

Thank you very much for your review

Reviewer 2 Report

Thank you for giving me the opportunity to review this paper. However, I have shared my viewpoints.

The abstract is well organized. However, if the author(s) can add the limitation and areas for future research at the end of the abstract part, it will be great.

You have written “Theoretical background and Hypothesis development”, but I did not find any theoretical assumptions!!!! You must discuss the assumptions of your used theory or theories, and then; you can formulate your hypotheses.

The theoretical framework that you drew is not correct. You should not use the co-variance sign between the independent and dependent variables.

In the methodology, there is confusion about the unit of analysis (who are your respondents or what is your sample). In the current form, the population and sample are making confusing.

Which sampling technique have you used to select your sample? You can read the following article for the sampling techniques.

.

Sampling Techniques (Probability) for Quantitative Social Science Researchers: A Conceptual Guidelines with Examples (https://sciendo.com/article/10.2478/seeur-2022-0023)

In the result section, please re-check Table 3. The reliability value is 1.00. Besides, the AVE is also 1.00. It is confusing.

You have written the discussion and conclusion in the same heading. Please, separate these sections in two.

The discussion part needs to improve. Please add more recent findings in relation to your existing finding.

Before going to the conclusion, you are requested to add the “contribution or implication” of the study. 

Author Response

“The abstract is well organized. However, if the author(s) can add the limitation and areas for future research at the end of the abstract part, it will be great.”

Thank you very much for your comment, based on your recommendation the limitations and future lines of research have been introduced in the abstract.

“You have written “Theoretical background and Hypothesis development”, but I did not find any theoretical assumptions!!!! You must discuss the assumptions of your used theory or theories, and then; you can formulate your hypotheses.”

Thank you very much for your recommendation, it has certainly helped us to improve the quality of the paper. We have strengthened the review of the literature, explaining which are our theoretical assumptions that allow us to formulate the proposed hypotheses.

“The theoretical framework that you drew is not correct. You should not use the co-variance sign between the independent and dependent variables.”

Thank you very much for your recommendation. Following your comment, we have eliminated the sign of the covariance between the analyzed variables, which is reflected in the new hypotheses.

“In the methodology, there is confusion about the unit of analysis (who are your respondents or what is your sample). In the current form, the population and sample are making confusing.”

Thank you very much for your comment. To avoid confusion in the methodological section, we have specified that the unit of analysis are the wineries, so that each observation represents a winery.

“Which sampling technique have you used to select your sample? You can read the following article for the sampling techniques. Sampling Techniques (Probability) for Quantitative Social Science Researchers: A Conceptual Guidelines with Examples (https://sciendo.com/article/10.2478/seeur-2022-0023)”

Thank you for sharing the publication with us. It was not necessary to use sampling techniques for the study, given that, as specified, the sample coincides with the population under study (the wineries belonging to the Ribera del Duero wine-growing territory). What we have had to identify are the companies in the population, which we have done through the SABI database, as explained in the methodological section.

“In the result section, please re-check Table 3. The reliability value is 1.00. Besides, the AVE is also 1.00. It is confusing.”

Thank you very much for your comment, it has undoubtedly helped us to improve the quality and facilitate the reading for potential readers. The composite reliability values of Cronbach's Alpha and the AVE have a value of 1 for both the wine tourism variable and the organic agriculture variable. This is due to the fact that both constructs are composed of a single indicator. We have added this in the revised version to facilitate understanding of the results.

“You have written the discussion and conclusion in the same heading. Please, separate these sections in two.”

Thank you very much for your recommendation, based on your suggestion we have separated the heading into two.

“The discussion part needs to improve. Please add more recent findings in relation to your existing finding.”

Thank you very much for your recommendation, based on your suggestion we have deepened the discussion section, introducing the latest research that has addressed our research objective.

“Before going to the conclusion, you are requested to add the “contribution or implication” of the study.”

Thank you very much for your recommendation, based on your suggestion we have introduced a section on theoretical and practical implications of the study so that they can be properly identified in the manuscript.

Round 2

Reviewer 2 Report

In the abstract and methodology part, if you can add the sampling technique, it will be good work.

Special thanks to all of the authors to cover all of the given corrections. 

Best of luck.  

Author Response

"In the abstract and methodology part, if you can add the sampling technique, it will be good work. Special thanks to all of the authors to cover all of the given corrections. Best of luck." 

Thank you very much for your work in the review, it has helped us a lot to improve the quality of the paper.

In our research, the sample and the population coincide, since we have used data from the SABI database to access the 263 wineries that make up the population (all wineries belonging to the Ribera del Duero wine region). We have added this in the new version, both in the abstract and in the methodological section, to facilitate the understanding of the research for future readers.

Thank you again for your recommendations.